# Identification of a Novel Biosurfactant with Antimicrobial Activity Produced by *Rhodococcus opacus* R7

**DOI:** 10.3390/microorganisms10020475

**Published:** 2022-02-21

**Authors:** Jessica Zampolli, Alessandra De Giani, Alessandra Di Canito, Guido Sello, Patrizia Di Gennaro

**Affiliations:** 1Department of Biotechnology and Biosciences, University of Milano-Bicocca, Piazza della Scienza 2, 20126 Milan, Italy; jessica.zampolli@unimib.it (J.Z.); alessandra.degiani@unimib.it (A.D.G.); alessandra.dicanito@unimib.it (A.D.C.); 2Department of Chemistry, University of Milano, via Golgi 19, 20133 Milan, Italy; guido.sello@unimi.it

**Keywords:** *Rhodococcus*, biosurfactants, antimicrobials, peptides, biosynthesis, biodegradation

## Abstract

*Rhodococcus* members excrete secondary metabolites, especially compounds which act as biosurfactants. In this work, we demonstrated the ability of *Rhodococcus opacus* R7 to produce a novel bioactive compound belonging to the class of biosurfactants with antimicrobial properties during the growth on naphthalene. Chemical and biochemical analyses of the isolated compound demonstrated that the biosurfactant could be classified as a hydrophobic peptide. The ESI-full mass spectrometry revealed that the isolated biosurfactant showed a molecular weight of 1292 Da and NMR spectra evidenced the composition of the following amino acid residues: Ala, Thr, Asp, Gly, Ser. Surfactant activity of the *R*. *opacus* R7 compound was quantified by the critical micelle dilution (CMD) method and the critical micelle concentration (CMC) was estimated around 20 mg L^−1^ with a corresponding surface tension of 48 mN m^−1^. Moreover, biological assays demonstrated that *R*. *opacus* R7 biosurfactant peptide exhibited antimicrobial activity against *Escherichia coli* ATCC 29522 and *Staphylococcus aureus* ATCC 6538 with the minimum inhibition growth concentration (MIC) values of 2.6 mg mL^−1^ and 1.7 mg mL^−1^, respectively. In this study for the first time, a hydrophobic peptide with both biosurfactant and antimicrobial activity was isolated from a bacterium belonging to *Rhodococcus* genus.

## 1. Introduction

*Rhodococcus* genus comprises varied microorganisms taxonomically associated with the Actinobacteria that possess an extensive set of catalytic enzymes revealing their remarkable catabolic versatility, great biodegradation potential, and the robustness to thrive in harsh environments [1,2]. Moreover, an additional feature of *Rhodococcus* members is the biosynthetic capacity in producing secondary metabolites, including biologically active compounds which act as biosurfactants [3]. Biosurfactants are functional amphipathic compounds, often with hydrophilic and hydrophobic residues that can be secreted or remain adherent to the cell surface.

Generally, bacteria produce bioactive molecules under different conditions: when exposed to hydrophobic (water-insoluble) carbon sources favoring their bioavailability and uptake, or in the presence of limiting growth conditions, or as protection against stress environmental conditions [4,5]. Some bacteria produce compounds with surfactant activity in response to water-soluble compounds [6].

Biosurfactants show diversity in structural composition and intriguing characteristics including lower surface tension, high foaming, high biodegradability, low toxicity, and self-assembly properties that make them environmentally friendly [7,8]. Furthermore, among their most interesting features, high efficiency and lower dosage requirement are often mentioned [9]. Accordingly, biosurfactants can be used in a broad range of applications and industries, such as food and textile industries, pharmaceuticals and cosmetics, oil recovery and bioremediation for heavy metals, oil spills, and other pollutants’ removal [5,7,9].

Surface-active molecules produced by Actinobacteria are categorized in several chemical groups; most of them are glyco-derivatives, including trehalose glycolipids, oligosaccharides (also as macrocycles, and terpene containing derivatives), and glycans. Furthermore, several lipopeptides, both cyclic and acyclic, or hydrophobic peptides are reported [5]; for example, *Streptomyces tendae* Tuë 901/8c produces an extracellular biosurfactant belonging to the peptide group with the ability to decrease the surface tension of water [10].

The main biosurfactants produced by the *Rhodococcus* genus belong to the class of low-molecular-weight glycolipids; only a few members of the genus are known to produce lipopeptides [2,11,12], and none of the members are known for their ability to produce hydrophobic peptides. Despite this, the development of bioinformatics tools and the increasing amount of available *Rhodococcus* genomes have also led to the prediction of a diverse number of biosynthetic gene clusters (BGCs) harbored by *Rhodococcus* strains, including a high number of nonribosomal peptide synthetase (NRPS)-encoding BGCs. NRPS are multi-modular enzyme complexes comprising the activation or adenylation domain (A), modification or a peptidyl carrier protein (PCP) or thiolation domain (T), and condensation domain (C). A C domain subtype called the C-starter domain is usually the first in the assembly domain lines that drive the acylation of the first amino acid (with the presence of β-hydroxy or β-amino fatty acid) of the peptide moiety [1]. In general, lipopeptides or hydrophobic peptides from all bacteria are very interesting because of their high activity in lowering surface tension [9,10,12,13,14,15] and because they often present antimicrobial activity [3,16].

Due to increase in antibiotic-resistant microorganisms, there is a high demand for new antimicrobial agents and the combination of surfactant properties with antimicrobial activity could facilitate their application as alternative antibiotics [17].

Several papers showed that biosurfactants produced by different bacteria exhibit several activities including antimicrobial, anti-adhesive, antiviral, anticancer, anti-HIV, anti-inflammatory as well as immunomodulatory activities [8,13,18].

The first detailed studies of antimicrobial properties of novel biosurfactant commenced over the last two decades. In particular, the lipopeptide or hydrophobic peptide class is of particular interest for its manifold attractive properties, such as antibiotic activity [19,20].

Among the first studies, well-characterized lipopeptides derived from *Bacillus* strains [17] and, in detail, the surfactin produced by *B*. *subtilis* is able to inhibit the growth of *Mycoplasma* spp. involved in the infectious disease of the urinary tract [6,21]. The most studied lipopeptides from *Bacillus* strains can be included in one of the following classes: surfactins, iturins, and fengycins [22], and the well-known hydrophobic peptide from *Streptomyces tendae* can be included in the streptofactin class. Different lipopeptides or hydrophobic peptides show a broad antimicrobial spectrum in terms of both antagonist strains and MIC values against reference and pathogenic microorganisms such as *Escherichia coli* and *Staphylococcus aureus* [23,24,25].

Many peptides with antimicrobial activity, containing a high molar ratio of unusual amino acids called 2,4-diaminobutyric acid (DAB), were isolated from *Bacillus* and *Paenibacillus* strains, such as permeatin A, polypeptin A, B, and C, and polymyxin B [26].

Concerning the lipopeptides from *Rhodococcus,* none of those characterized showed antimicrobial activity, except for the mixture of lipopeptides from *R*. *ruber* that showed antifungal activity [27].

Among members of the *Rhodococcus* genus, *Rhodococcus opacus* strain R7 is known as a powerhouse of biodegradative functions, possessing a high number of gene clusters, and the largest genome with 10.1 Mb [28,29].

This study reports the ability of *R*. *opacus* R7 to produce a novel biosurfactant belonging to peptide group also exhibiting antimicrobial activity. The new biosurfactant was isolated and characterized, and the measure of water-surface tension and the antimicrobial activity against Gram-positive and Gram-negative microorganisms were determined. For the first time, a hydrophobic peptide with both surfactant and antimicrobial activity was isolated from a bacterium belonging to *Rhodococcus* genus.

## 2. Materials and Methods

### 2.1. Growth Conditions of the Biosurfactant-Producing Bacterial Strain

The microorganism used in this study is *R. opacus* R7 (deposited in the Collection Institute Pasteur CIP identification number 107348, [30]).

Biosurfactant production was carried out by culturing *R*. *opacus* R7 in a 3 L flask containing 500 mL M9 mineral medium [31] supplemented with naphthalene (1 g naphthalene crystals/L of culture) as the only carbon and energy source at 30 °C under shaking (120 rpm) up to 96 h corresponding to an optical density at 600 nm (O.D._600_) of around 2. Biosurfactant production and the subsequent extraction were accomplished after 48 h of *R*. *opacus* R7 cultivation.

### 2.2. Determination of Surface Activity and Surface Tension

The surface activity of the cell-free supernatant (CFS) deriving from *R*. *opacus* R7 growth on M9 mineral medium supplemented with naphthalene, the derived crude biosurfactant extract, and the purified biosurfactant compound were preliminarily measured by the oil spreading assay (water was used as a negative control) [32]. An amount of 100 µL of each sample were gently added onto the center of 700 µL diesel fuel film previously deposited onto the surface of around 20 mL of milliQ water in a Petri dish (90 mm in diameter). The diameter of the displaced circle was measured. The oil spreading assay on the CFS was performed sampling *R*. *opacus* R7 culture broth every 24 h of growth.

The surface tension (ST) of the CFS and the purified biosurfactant compound resuspended in water were measured with the Du Nouy ring method using K-8 tensiometer (Kruss, Hamburg, Germany) at room temperature [33]. The ST of *R*. *opacus* R7 CFS was recorded collecting 20 mL of the sample at regular intervals: 0, 6, 24, 48, 72, 96 h.

The critical micelle dilution (CMD) was determined for the purified biosurfactant compound from the ST measured and it corresponds to the measure of the dilution factor to reach the level of critical micelle concentration (CMC) [34].

### 2.3. Isolation and Purification of the Biosurfactant Compound

First, microbial cells were removed from the culture broth (500 mL) by centrifugation at 6000 rpm for 15 min (Eppendorf Inc., Hamburg, Germany). Then, the CFS was adjusted to pH 2.0 using HCl (6 M); in this condition the content of the CFS converts to a protonated form, favoring the recovery in an organic solvent. Before solvent extraction, water was removed by rotary evaporator at 40 °C under reduced pressure (Eppendorf Inc., Hamburg, Germany) and the residual solid was extracted with 30 mL methanol over-night. Finally, the obtained extract in methanol was separated by filtration. The final crude extract was obtained by solvent evaporation, and it was maintained dry at 4 °C. In order to evaluate the crude biosurfactant extract content, it was analyzed on silica gel TLC plates, developed in CH_3_Cl:Me(OH):H_2_O [40:51:9] or CH_3_Cl:Me(OH):H_2_O:CH_3_COOH [3:6:1:0.3] and visualized with iodine, UV, and ninhydrin.

Furthermore, the crude biosurfactant extract was purified by silica gel (230–400 mesh, Merck) column chromatography using step-wise elution with CH_3_Cl:Me(OH):H_2_O [40:51:9] at room temperature (modified method from Peng et al. [9]). The fractions were collected, and the purity of each fraction was checked by silica gel TLC plates developed as reported above. The fractions containing the same compounds were pooled together and tested for the biosurfactant activity after volume reduction.

### 2.4. Determination of Carbohydrates, Unsaturated Carbon Chains, and Oxidizable Components of the Biosurfactant Compound

Tests with ethanol–sulphuric acid, iodine vapor, UV, and cerium:molybdate reagent were performed on the purified biosurfactant to evaluate the presence of carbohydrates, unsaturated carbons, and oxidizable components. Each assay was carried out on TLC plates. The results were compared to the reaction of glucose and oleic acid to the same reagents.

### 2.5. Acid Hydrolysis of the Biosurfactant Compound

In order to evaluate the amino acid composition of the biosurfactant compound, 10 mg of the biosurfactant were resuspended in 4 mL HCl 5M and hydrolyzed at 121 °C in a 1000-watt microwave oven (Ethos Up, Milestone, Italy) with the following procedure: heating for 10 min to reach 121 °C and holding 121 °C for 10 min.

The hydrolyzed sample was then analyzed on TLC plates developed with CH_3_Cl:Me(OH):H_2_O:CH_3_COOH [3:6:1:0.3] and visualized with ninhydrin.

In particular, polar, apolar, acid, and neutral amino acids were detected in the biosurfactant sample compared to the pure amino acids belonging to the different categories used as references.

### 2.6. Effects of Proteinase K Enzyme on the Isolated Biosurfactant Compound and Total Protein Assay

The total protein concentration of the purified biosurfactant was assessed using Pierce BCA protein assay kit (Thermo Fisher Scientific, Monza, Italy) according to the manufacturer’s protocol and by the method of Bradford using Coomassie brilliant blue with bovine serum albumin as a standard [35]. Protein concentrations were calculated from the standard curve by 20 µg mL^−1^ bovine albumin serum.

In order to evaluate the extract compound sensitivity to the proteinase K (pH 7.5; Sigma Aldrich, Milano, Italy), 100 mg mL^−1^ of the purified biosurfactant were treated with proteinase K (ratio 1:4) at its optimal pH. After 3 h of incubation at 37 °C, the reaction was stopped at 4 °C. The compound without any heat or enzyme treatments was used as a control sample. The result was also compared with the effect of the compound incubated at 37 °C, the effect of the milliQ sterile water incubated at 37 °C, and the only milliQ sterile water as controls. The agar well diffusion assay was carried out to test the remaining activity against the indicator strains: *E.*
*coli* ATCC 25922 and *S.*
*aureus* ATCC 6538 [36].

### 2.7. ESI-Full Mass Spectrometry (MS) of the Biosurfactant Compound

The biosurfactant compound was analyzed by the electrospray ionization ESI-full mass spectrometry on a Q-TOF Synapt G2-Si mass spectrometer (Waters, Sesto San Giovanni, Italy). The MS was operated in positive and negative ionization mode with a capillary voltage of 2.8 kV in positive mode and 3.0 kV in negative mode and a sampling cone voltage of 30 V in both modes. The full-scan data were acquired from 50 to 2000 M/Z with a 0.2 s scan time and a 0.1 s inter scan delay over a 17.5 min run time, a de-solvation temperature of 350 °C, source temperature of 120 °C, cone gas flow of 50 L/h and de-solvation gas flow of 750 L/h.

### 2.8. NMR Spectroscopy of the Biosurfactant Compound

For the structure prediction of the purified biosurfactant, one- and two-dimensional NMR spectra were obtained by Bruker Avance 500 MHz and Avance I 600 MHz spectrometers (Karlsruhe, Germany) using deuterated CH3OD as the solvent, dissolving the biosurfactant at a concentration of 5 mg mL^−1^. NMR experiments were performed at 25 °C with a 5-mm z-gradient inverse broad-band probe equipped with a pulsed gradient unit capable of producing magnetic field pulse gradients, in the z-direction, of 53.5 G cm^−1^. Chemical shifts were expressed in parts per million (ppm). Different spectra were obtained: ^1^H, ^13^C, DEPT, HSQC.

### 2.9. Antimicrobial Activity

The crude biosurfactant extract and the purified biosurfactant were prepared in 100 mg mL^−1^ milliQ sterile water and stored in sterile glass vials. A modification of the protocol of Santini et al. [36] well diffusion agar assay was used. Antagonist selected bacteria *E*. *coli* ATCC 25922 and *S*. *aureus* ATCC 6538 were inoculated into LB medium and allowed to grow until the O.D._600_ was 0.5, corresponding approximately to 10^7^ CFU mL^−1^. Each culture was then diluted 1/10, 75 µL were inoculated into 20 mL of melted LB and then poured into plates to solidify. Three to five wells of 8 mm in diameter were made on each agar plate with a sterile cylinder and 100 μL of crude biosurfactant extract and purified biosurfactant were dispensed into each well. In order to evaluate the sensitivity to the proteinase K of the purified biosurfactant compound, 100 μL of both purified biosurfactant treated with proteinase K at 37 °C and the extract treated at 37 °C were dispensed into each well. Not treated biosurfactant compound, proteinase K in milliQ sterile water at the same concentration used to treat the biosurfactant, and milliQ sterile water (100 μL) were used as controls. Plates were incubated overnight at 37 °C in aerobiosis. The growth inhibition zones around the wells were measured [37].

The minimum inhibitory concentration (MIC) values were determined by 1/5 serially diluting the purified biosurfactant in milliQ water and using *E*. *coli* ATCC 25922 and *S*. *aureus* ATCC 6538 as indicator bacterial strains [38]. For each growth test well in LB medium, 10 μL of bacterial cells (10^6^ CFU mL^−1^ per well) were inoculated and water was used as a control. After incubation at 37 °C for 24 h, the optical density was measured at O.D._600_ using a microplate reader (Victor, ELx 800, Milano, Italy). A T-student test (two tailed) was applied for the growth inhibition of *E*. *coli* and *S*. *aureus* at MIC value to evaluate the significance of the results. Statistically significant differences were considered as *p*-value < 0.05 (at 95% confidence).

### 2.10. AntiSMASH Analysis of R. opacus R7

*R. opacus* R7 genome (GCF_000736435.1) was analyzed with antiSMASH using default settings (http://antismash.secondarymetabolites.org, accessed on 2 February 2022) and the antiSMASH-incorporated NaPDoS pipelines to identify the principal biosynthetic gene clusters (BGCs).

## 3. Results

### 3.1. R. opacus R7 Growth and Biosurfactant Production Profile

The biosurfactant production was evaluated cultivating *R. opacus* R7 in M9 mineral medium supplemented with 1 g L^−1^ naphthalene up to 96 h. *R*. *opacus* R7 growth was recorded at 6, 24, 48, 72, and 96 h, and the supernatant was collected at each time to measure the change of water-surface tension.

The surfactant activity detection was preliminarily evaluated through the oil spreading assay testing the cell-free supernatant (CFS) of *R*. *opacus* R7 culture every 24 h. The test indicated that the spreading of the oil due to the presence of *R*. *opacus* R7 CFS was qualitatively stable from 48 h upward and Figure 1 shows the result of the test at 48 h.

In order to precisely quantify the biosurfactant activity, the ST of the CFS was measured at different time-points of *R*. *opacus* R7 growth on naphthalene (Figure 2). The ST decrease during the time-growth (at the beginning time was equal to 70 mN m^−1^) and the lowest ST (48 mN m^−1^) was detected at 48 h of *R*. *opacus* R7 growth (middle–late exponential phase) that remained stable until the end of the stationary phase. This result shows the biosurfactant trend production indicating that the biosurfactant production was dependent on the growth and confirming the preliminary oil spreading assay.

Altogether, these results indicated that the CFS from *R*. *opacus* R7 culture on naphthalene has biosurfactant properties suggesting that it contained a biosurfactant compound able to reduce the ST.

### 3.2. Isolation and Identification of the Biosurfactant Compound

On the basis of the previous results, a protocol was set up to isolate and recover the biosurfactant compound from *R*. *opacus* R7 cell-free supernatant after 48 h of growth.

The CFS of *R*. *opacus* R7 culture was acidified, dried, solvent extracted, and purified using silica gel column chromatography. The subsequent eluted fractions of the crude extract were examined by running on TLC with the same solvent using UV, iodine, and ninhydrin. These fractions were pooled and concentrated for further characterization.

Figure 3 shows the result of the purification process with respect to the unpurified mixture. Figure 3 Panel A qualitatively shows a TLC detected with ninhydrin containing multiple spots of the unpurified mixture; Figure 3 Panel B shows a TLC detected with ninhydrin highlighting a single spot that indicates the purification of the crude extract isolated from *R*. *opacus* R7 CFS.

The surface activity of both the crude extract and the purified compound was preliminarily tested by the oil spreading assay to evaluate that the biosurfactant compound was present in each step of the extraction and the purification processes (Figure 4A,B). This assay evidenced that the compound was able to spread the oil at every step of the purification demonstrating its biosurfactant property.

### 3.3. Characterization of the Isolated Biosurfactant Compound

Initially, the isolated and purified biosurfactant compound was characterized by several properties.

The preliminary test with ninhydrin was used to evaluate the presence of amines or α-amino acids and the purified biosurfactant compound reacted positively to the reactive revealing that contains amino acid residues (Figure 3).

Other qualitative tests developed with ethanol–sulphuric acid, iodine vapor, and the cerium:molybdate reagent were performed on the purified biosurfactant to evaluate the presence of carbohydrates, unsaturated carbon chains, and oxidizable components, respectively. Comparing the results to glucose and oleic acid reaction, the purified biosurfactant showed a negative response to the three assays. These indications suggested that the isolated compound is not composed of sugars, unsaturated carbons, and oxidizable components.

In order to evaluate the presence of amino acidic components, the acid hydrolysis was performed on 10 mg of the biosurfactant compound in the presence of HCl 5 M in a microwave oven that ensured a rapid and effective reaction [39].

The outcome of the hydrolysis was visualized on TLC developed with ninhydrin. Figure 5 shows that the hydrolyzed biosurfactant compound results in at least five different spots with different R_fs_ compared with the not hydrolyzed biosurfactant compound. This experimental procedure confirmed the presence of amino acid residues observed with the preliminary test with ninhydrin. In particular, polar, apolar, acid, and neutral amino acids were detected in the biosurfactant sample compared to the pure amino acids belonging to the different categories used as references.

### 3.4. Effects of Proteinase K Enzyme on the Isolated Biosurfactant Compound and Total Protein Assay

Based on the preliminary results and the positive response to both BCA and Bradford protein assays on the purified biosurfactant compound, the peptide component of the isolated compound was tested to evaluate its essential and functional role. Thus, the sensitivity of the purified biosurfactant compound was tested in the presence of proteinase K after incubation at 37 °C with the hydrolytic enzyme. As it is known that biosurfactants with a peptide nature show also antimicrobial activity, the effect after the treatment with proteinase K by the well agar diffusion assay antimicrobial test was evaluated. The treatment with proteinase K showed that the antimicrobial activity of the purified biosurfactant compound against *E*. *coli* ATCC 25922 and *S*. *aureus* ATCC 6538 was weakened by the treatment, while it was maintained in the sample not treated with proteinase K (Figure 6). These results demonstrated that the biosurfactant property of the isolated compound could be attributed mainly to the amino acid nature of the molecule and that the isolated compound also possesses an antimicrobial activity against other microorganisms.

### 3.5. Chemical Characterization of the Purified Biosurfactant

The purified biosurfactant was chemically characterized by Mass Spectrometry and NMR Spectroscopy. The ESI-full MS analysis on the purified biosurfactant revealed a compound with a molecular weight of 1292 Da (calculated MS (M^+^ - H_2_O): 1292 Da) (Appendix A).

The NMR spectra revealed the composition of the biosurfactant constituted by Ala, Gly, Asp, Thr, Ser. Of course, multiple residues in the sequence were expected (Appendix A and Figure 7). From the spectra, the following amino acids were identified: alanine (^1^H, 1.35, 3.75; ^13^C, 19.88, 48.45); aspartic acid (^1^H, 2.9, 3.05, 3.65; ^13^C, 37.05, 42.64, 53.72); glycine (^1^H, 3.35; ^13^C, 40.77); serine (^1^H, 3.6, 3.7; ^13^C, 37.05, 42.64); threonine (^1^H, 1.2, 3.4, 3.6; ^13^C, 19.88, 49.96, 53.72). No other components were identified in the isolated compound.

### 3.6. Activities of the Biosurfactant Compound

On the basis of the obtained results associated with the chemical nature and the behavior of the isolated compound, the isolated compound was classified within the class of biosurfactant peptides. To deeply characterize the isolated peptide from *R*. *opacus* R7, some properties of the isolated compound were measured in terms of surfactant activity and antimicrobial activity.

#### 3.6.1. Antimicrobial Activity

As it is known that some peptides possess antimicrobial activity, the activity spectrum of the biosurfactant isolated from *R*. *opacus* R7 was evaluated.

In order to monitor the biosurfactant purification process, the purified biosurfactant compound was screened for the antimicrobial activity against the most representative antagonists belonging to Gram-negative and Gram-positive bacteria such as *E*. *coli* ATCC 29522 and *S*. *aureus* ATCC 6538, and this activity was compared to the activity of the crude extract. The well diffusion agar assay demonstrated that the crude extract exerted antimicrobial activity and provided preliminary information about the measure of the halo inhibition of the growth. The purified biosurfactant was also tested by the well diffusion agar assay to confirm that this activity was maintained during the purification process. Results showed an inhibitory action of the biosurfactant compound with a concentration of 100 mg mL^−1^ against the two selected pathogens with a halo of inhibition of 2.6 cm and 2.7 cm, respectively.

Thus, the minimum inhibition growth concentration (MIC) value was evaluated. The MIC values of *R*. *opacus* R7 purified biosurfactant were 2.6 mg mL^−1^ and 1.7 mg mL^−1^, respectively, against *E*. *coli* ATCC 29522 and *S*. *aureus* ATCC 6538 as reported in Table 1. A T-student test was applied for the growth inhibition of *E*. *coli* and *S*. *aureus* at an MIC value showing a *p*-value < 0.05.

#### 3.6.2. Surfactant Activity Evaluated by the Critical Micelle Dilution (CMD)

The CMD is the measure of the concentration corresponding to the dilution factor to reach the level of critical micelle concentration (CMC); it derives from ST measurement of the purified surfactant that was diluted by different factors. Figure 8 reports the graph with the ST measured at different concentrations (dilutions in water). Results indicated that the estimated CMC is 20 mg L^−1^ and the corresponding ST is 48 mN m^−1^.

### 3.7. AntiSMASH Analysis of R. opacus R7

The prediction of secondary metabolites was performed on *R*. *opacus* R7 genome using antiSMASH. Different BGCs were identified such as polyketide synthase (PKS), terpene, saccharides, post-translationally modified peptides (RiPPs) and several non-ribosomal peptide synthetase (NRPS), classified in 32 known families. The ratio NRPS vs. PKS is high and among PKS types, only PKS type I was observed.

## 4. Discussion

*R. opacus* strain R7 is a metabolically versatile microorganism known for its ability to grow on several long- and medium-chain *n*-alkanes, carboxylic acids, and naphthenic acids, aromatic hydrocarbons belonging to the BTEX group (benzene, toluene, ethylbenzene, and xylenes) and polycyclic aromatic hydrocarbons (PAHs) such as naphthalene [29,40,41,42].

Considering the extraordinary ability of the *R*. *opacus* R7 strain in degrading different environmental contaminants, in this work the *R*. *opacus* R7 strain’s ability to produce a novel bioactive compound belonging to the class of biosurfactants with antimicrobial properties was demonstrated. It is noteworthy that *R*. *opacus* R7 produced this useful compound exploiting an environmental contaminant such as naphthalene [43]. Indeed, *R*. *opacus* R7 grown on naphthalene, as only carbon and an energy source, produced a correspondent CFS exhibiting biosurfactant properties detectable by oil spreading assay. Accordingly, the ST values of the CFS decreased during *R*. *opacus* R7 growth and reached the minimum level of 48 mN m^−1^. This value in relative accordance with the ST recorded for the CFS from *Paenibacillus* sp. D9 grew on diesel fuel (the maximum value reported was 43.3 mN m^−1^) [44]. The profile of *R*. *opacus* R7 biosurfactant production as a function of time indicated that the biosurfactant secretion was growth-dependent and the maximum production occurred at the middle-late logarithmic growth phase (around 48 h) at the same time point when the naphthalene crystals started disappearing. Such behavior is in accordance with the literature reporting that the limitation of essential medium components; for instance, nitrogen sources or the limiting availability of the carbon source can generate stress on the cells, and they can play a fundamental role for the biosurfactant production [45].

Consequently, a specific experimental procedure to extract a compound with stable surfactant activity from *R*. *opacus* R7 supernatant was developed to isolate and to recover biosurfactants. After acidification and drying of *R*. *opacus* R7 CFS, a solid-solvent extraction was performed with methanol. The purification was established after trying several conditions and it was performed with CH3Cl:Me(OH):H_2_O. The final procedure produced one compound visible on TLC showing a single spot.

Thus, the purified compound was then characterized for its chemical, biochemical, and biological properties. Notwithstanding the glycolipid class is the most extensively reported among the biosurfactant categories deriving from *Rhodococcus* spp. strains, the *R*. *opacus* R7 biosurfactant reacted negatively to the carbohydrate test as well as to the presence of unsaturated carbon chains and oxidizable components. On the contrary, the isolated compound reacted positively to ninhydrin suggesting the occurrence of amino acid residues. On this basis, the presence of peptide components was evaluated by acid hydrolysis of the biosurfactant compound using an efficient sealed system, such as a microwave oven. The hydrolysis results, visualized on TLC, confirmed the presence of amino acids belonging to polar, apolar, acid, and neutral amino acids compared to the not hydrolyzed biosurfactant. The validation of these results was obtained by measuring the total protein concentration and evaluating the effect of proteinase K treatment. In fact, the activity of the biosurfactant compound was lost after the treatment with the hydrolytic enzyme.

The chemical characterization of the isolated biosurfactant revealed that the mass spectra assign a molecular weight (1292 Da) that is in the range of small biosurfactants structurally related to peptides (as lipopeptides and hydrophobic peptides) isolated from other bacteria [10,14,25,46]. The NMR spectra confirmed the presence of amino acids residues including aliphatic hydrophobic and hydrophilic amino acids: Ala, Gly, Asp, Thr, and Ser, and the analysis excluded the presence of a lipophilic component in the isolated compound. Based on these chemical analyses, we can hypothesize that the structure of the biosurfactant compound is only made of amino acids, and this led to defining the *R*. *opacus* R7 molecule as a biosurfactant peptide. This differs from the most biosurfactants such as lipopeptides, because they contain a lipophilic moiety (e.g., an aliphatic acid or an alkyl derivative); however, this is not a strict requirement [20,47,48,49], as is demonstrated by MBSP1 which is active as a single peptide [15] and the hydrophobic peptide streptofactin from *Streptomyces tendae* [10].

Among the amino acids identified, Ala, Asp and Gly are worth mentionign because they are found in several reported lipopeptides and hydrophobic peptides. These amino acids could contribute to enhancing the hydrophobic activity of the compound. It is also known that short peptides can aggregate and assume special conformations when they are in the presence of lipophilic solvents; this is nicely demonstrated by the biosurfactant that shows its amphiphilic property when in touch with CHCl_3_ solvent-forming visible micelles [20].

These experimental data were consistent with the genetic repertoire of *R*. *opacus* R7. Indeed, in line with what was observed for other rhodococci [1], *R*. *opacus* R7 possesses the highest number of putative BGCs among 20 strains belonging to the *Rhodococcus* genus and a higher ratio NRPS vs. PKS compared to other actinomycetes. All NRPS identified in *R*. *opacus* R7 showed a C-starter domain belonging to the C domain subtype that drives the acylation of the first amino acid of the peptide moiety.

Although molecules with both biosurfactant activity and peptide components exerting antimicrobial activity are often reported [5,50,51], among the few lipopeptides isolated from *Rhodococcus* spp. strains, only the work by Yalaoui-Guellal and coworkers [27] reported a biosurfactant from *Rhodococcus ruber* with a high antifungal activity that was obtained using the clear zone spot test without a quantification [27]. Interestingly, the biosurfactant compound isolated from *R*. *opacus* R7 showed a broad range of antimicrobial activity towards Gram-positive and Gram-negative bacteria. The MIC values of *R*. *opacus* R7 purified biosurfactant were comparable to literature data [23,52].

The surfactant activity of *R*. *opacus* R7 purified compound was also quantified by the CMD method that allowed estimation of the CMC around 20 mg L^−1^ with a corresponding ST of 48 mN m^−1^.

Comparing the surfactant property of the isolated compound from the *R*. *opacus* R7 strain to other biosurfactants from *Rhodococcus*, i.e., the lipopeptide produced by *Rhodococcus* sp. TW53 [9], the *R*. *opacus* R7 biosurfactant showed a higher ST (48 mN m^−1^) at the maximum level of its naphthalene metabolism. This can be explained by the fact that *R*. *opacus* R7 grew in an atmosphere saturated with this PAH, and the strain does not need direct contact with the molecule. Rather, *R*. *opacus* R7 growth on naphthalene undergoes an oxidative stress effect that generates the production of bioactive compounds, such as the production of a hydrophobic peptide [53].

## 5. Conclusions

In conclusion, the biosynthetic capacity of valuable compounds is an additional benefit that bacteria belonging to *Rhodococcus* genus can exploit, meanwhile, to the advantage of degrading contaminants such as naphthalene. Thereby, strains of *Rhodococcus* genus represent a rich source of structurally complex bioactive agents that can play an important role in several fields such as biosurfactants and antimicrobials for food industries, pharmaceuticals, cosmetics, oil recovery, and pollutant removal.

This work provides the identification and characterization of a new *R*. *opacus* R7 biosurfactant peptide that possesses also antimicrobial activity that could be leveraged as a promising candidate for biomedical and biotechnological applications.

## Figures and Tables

**Figure 1 microorganisms-10-00475-f001:**
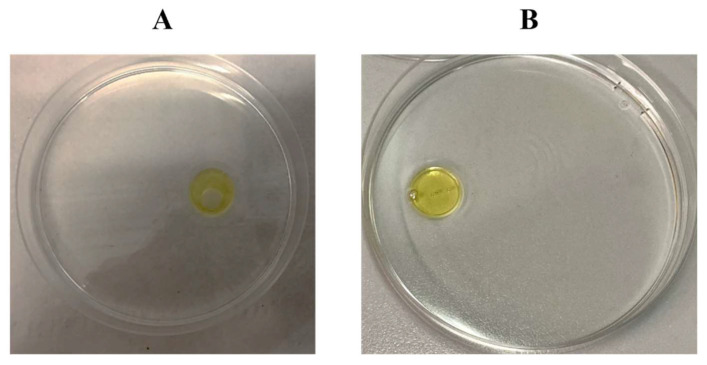
Oil spreading assay to evaluate the surface activity of the CFS produced by *R*. *opacus* R7 growth on M9 mineral medium supplemented with naphthalene for 48 h (**A**) and the control with water (**B**).

**Figure 2 microorganisms-10-00475-f002:**
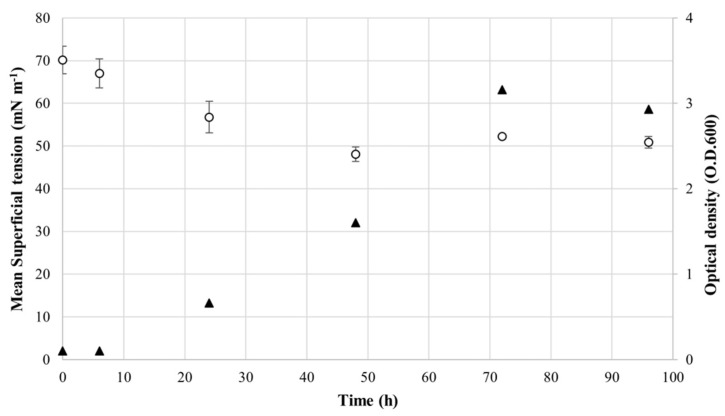
Analysis of the surface tension of CFS from *R*. *opacus* R7 grown on mineral medium supplemented with naphthalene. *R*. *opacus* R7 growth was measured by optical density (O.D._600_) (black triangles) and the surface tension values of CFS were obtained by tensiometer analysis (white circles). Mean values are presented obtained through triplicate treatments.

**Figure 3 microorganisms-10-00475-f003:**
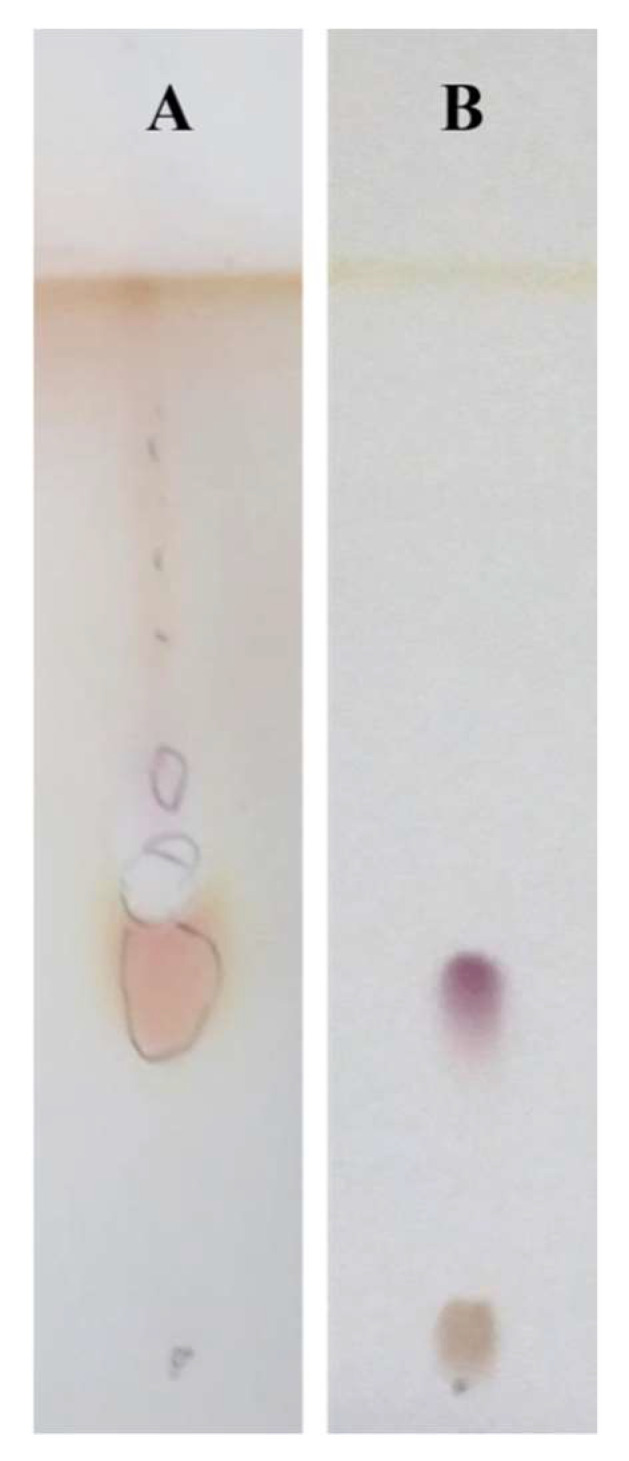
Thin layer chromatography analysis of the unpurified biosurfactant compound (**A**) compared to the purified biosurfactant (**B**) produced by *R*. *opacus* R7 developed with CH3Cl:Me(OH):H_2_O [40:51:9] detected with ninhydrin.

**Figure 4 microorganisms-10-00475-f004:**
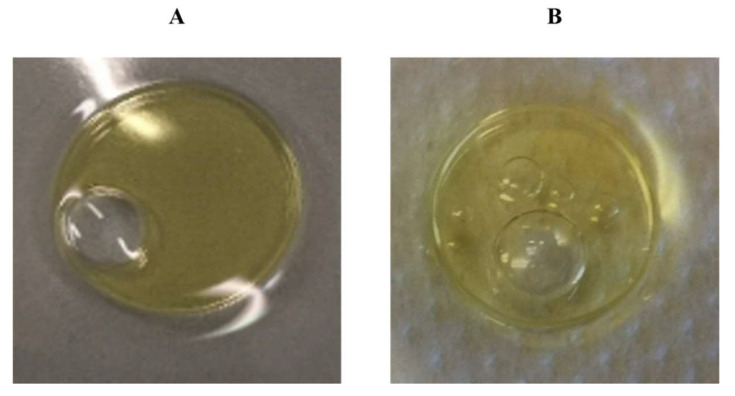
Oil spreading assay to evaluate the surface activity of the unpurified biosurfactant compound (**A**) and the purified biosurfactant (**B**) isolated from *R*. *opacus* R7 during growth on M9 mineral medium supplemented with naphthalene at 48 h.

**Figure 5 microorganisms-10-00475-f005:**
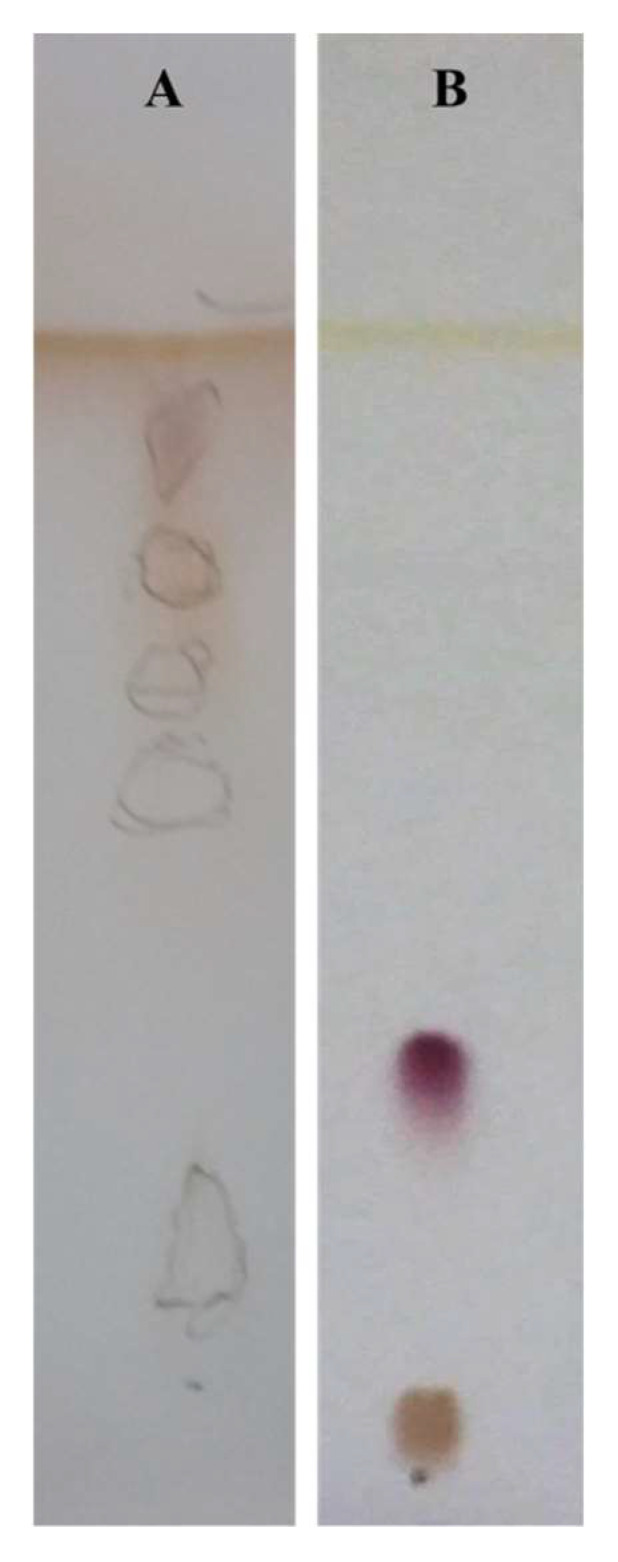
Thin layer chromatography analysis of the hydrolyzed biosurfactant (**A**) and the not-hydrolyzed biosurfactant (**B**) produced by *R*. *opacus* R7 developed with CH_3_Cl:Me(OH):H_2_O:CH_3_COOH [3:6:1:0.3] detected with ninhydrin.

**Figure 6 microorganisms-10-00475-f006:**
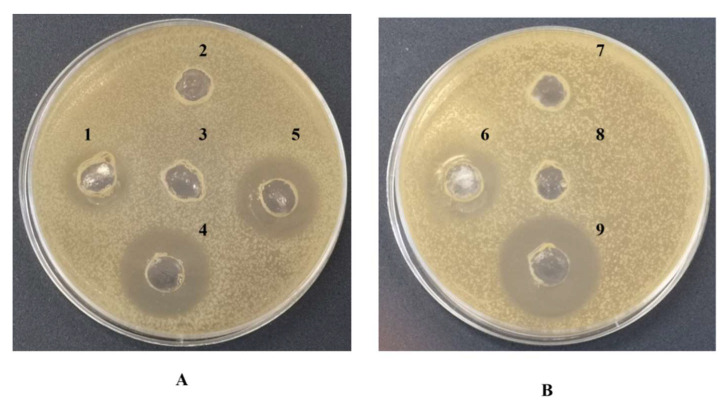
Sensitivity test of the purified biosurfactant from *R*. *opacus* R7 to proteinase K. Panel (**A**) shows the antimicrobial activity against *E*. *coli* ATCC 25922 of the purified biosurfactant treated the proteinase K (1), sterile milliQ water treated with proteinase K (2), sterile milliQ water not treated with proteinase K and not incubated (3), the purified biosurfactant not treated with proteinase K (4), the purified biosurfactant not treated with proteinase K incubated at 37 °C (5). Panel (**B**) shows the antimicrobial activity against *S. aureus* ATCC 6538 of the purified biosurfactant treated the proteinase K (6), sterile milliQ water treated with proteinase K (7), sterile milliQ water not treated with proteinase K and not incubated (8), the purified biosurfactant not treated with proteinase K (9).

**Figure 7 microorganisms-10-00475-f007:**
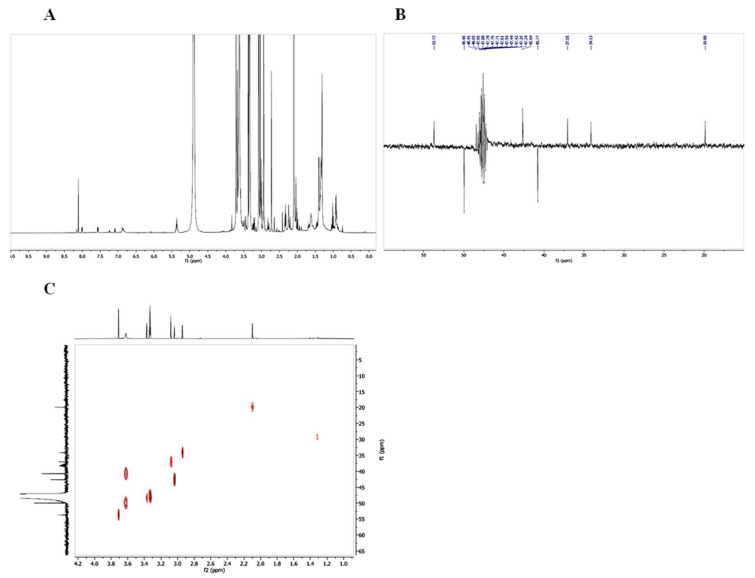
The NMR spectra of the biosurfactant compound. 1H in CD3OD (10 to 0 ppm) (**A**); 13C DEPT-135 in CD3OD (60 to 15 ppm) (**B**); 1H-13C HSQC one bond in CD3OD (**C**).

**Figure 8 microorganisms-10-00475-f008:**
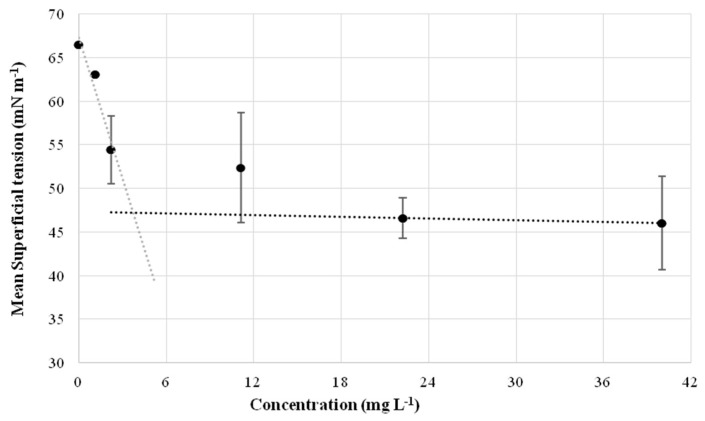
Analysis of surface tension and calculation of critical micelle dilution (CMD) of the purified biosurfactant compound extracted from *R*. *opacus* R7 CFS after 48 h of growth on naphthalene. The black dots represent the measured surface tension of the purified biosurfactant at different concentrations (dilutions in water). The grey dotted line represents the biosurfactant monomers while the black dotted line represents the biosurfactant micelles. Their intersection represents CMD, the estimated CMC is around 20 mg L^−1^.

**Table 1 microorganisms-10-00475-t001:** MIC spectrum of the purified biosurfactant produced by *R*. *opacus* R7 against *E*. *coli* ATCC 29522 and *S*. *aureus* ATCC 6538.

Biosurfactant Concentration mg mL^−1^	*E*. *coli* ATCC 25922 Growth Inhibition (%)	*S*. *aureus* ATCC 6538 Growth Inhibition (%)
100.0	100	100
66.7	98	100
44.4	100	100
29.6	100	100
19.8	100	100
13.2	100	100
8.8	100	100
5.9	100	100
3.9	100	100
2.6	30	100
1.7	8	53
1.2	2	0

## Data Availability

All data generated during this study are included in this article.

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
