# Peer review of "Identification of a Novel Biosurfactant with Antimicrobial Activity Produced by Rhodococcus opacus R7"

_microorganisms, 2022, doi:10.3390/microorganisms10020475_

Round 1

Reviewer 1 Report

First of all, I would like to congratulate the authors of this article, as I think they have done a great job.

The article describes the biosynthesis of a biosurfactant with antimicrobial capacity by Rhodococcus opacus R7 generated from a potential contaminant such as naphthalene. This article is highly relevant, since it addresses current needs such as the removal of contaminants and the production of new antimicrobial compounds.

The content of the article is exposed in a clear way and many relevant results are presented. The original findings of this article represent a valuable contribution to the field of microbial biotechnology. The article is perfectly aligned with the aims and scope of Microorganisms.

In reference to the use of English, the style is appropriate and understandable. Perhaps there are too many connectors at the beginning of sentences that could be omitted. I suggest that some of them could be removed when they do not make substantial contributions to the text.

In reference to specific text changes, I propose the following:

Introduction

  • In text “…they often present an antimicrobial activity”. Remove “an”.
  • In text “…facilitate their application as as alternative antibiotics”. Remove one “as”.
  • In text “Several papers showed that biosurfactants produced by different bacteria exhibit several activities including antimicrobial, antiadhesive, antiviral, anticancer, anti-HIV, anti-inflammatory as well as immunomodulatory activities [8,13,18]. Remove last “activities”. Perhaps this paragraph corresponds more to the "Discussion" section.
  • In text “Well-known hydrophobic peptide from S. tendae…”. Change S from red to black font colour.
  • In text: “lipopeptides from ruber that showed an antifungal activity”. Remove “an”.
  • In text: “peptides also exhibiting an antimicrobial activity”. Remove “an”.

Materials and Methods

  • In title 2.1: “Growth condition of the biosurfactant-producing bacterial strain”. Replace “condition” with “conditions”.
  • In text: “carried out by culturing of opacus R7 in a 3-L flask”. Remove “of”. Remove hyphen in 3-L.
  • In text: “However, the biosurfactant production…”. Remove “However, the”.
  • In text: “One hundred μL of each sample were…”. Replace “One hundred” with “100”.
  • In section 2.4: “3. Tests with ethanol-sulphuric acid, iodine vapor, UV, and cerium:molybdate reagent were performed on the purified biosurfactant to evaluate the presence of carbohydrates, unsaturated carbons, and oxidizable components, respectively. Each assay was carried out on TLC plates. The results were compared to the reaction of glucose and oleic acid to the same reagents.” The text that should go in this section is formatted as a new section title. This causes all subsequent numbering to be unbalanced. You should put this text as normal body text and correct the subsequent numbering.
  • In text: “biosurfactant at a concentration of 2-5 mg mL-1”. Does this mean that concentrations between 2 and 5 mg mL-1 were tested, or did it perhaps mean that a single concentration at 2.5 mg mL-1 was used? Change accordingly.
  • In text: “…ATCC 6538 were inoculated into the suitable medium”. Specify the culture medium used.

Results

  • In sections 4.4, 4.5 and 4.6 the first-person plural "we" is used repeatedly. These phrases should be changed to impersonal style.

Discussion

  • In text: “Rhodococcus opacus strain R7 is a…”. Replace “Rhodococcus with R.

References

  • A full stop must be added after each reference.
  • There are references, such as 15, 39 and 44, that have their doi underlined. Remove the underline.
  • References 2 and 31 do not have the year in bold.

Author Response

POINT BY POINT TO

REVIEWER 1

The article describes the biosynthesis of a biosurfactant with antimicrobial capacity by Rhodococcus opacus R7 generated from a potential contaminant such as naphthalene. This article is highly relevant, since it addresses current needs such as the removal of contaminants and the production of new antimicrobial compounds.

The content of the article is exposed in a clear way and many relevant results are presented. The original findings of this article represent a valuable contribution to the field of microbial biotechnology. The article is perfectly aligned with the aims and scope of Microorganisms.

In reference to the use of English, the style is appropriate and understandable. Perhaps there are too many connectors at the beginning of sentences that could be omitted. I suggest that some of them could be removed when they do not make substantial contributions to the text.

In reference to specific text changes, I propose the following:

Introduction

  • In text “…they often present an antimicrobial activity”. Remove “an”.
  • Reply: done
  • In text “…facilitate their application as as alternative antibiotics”. Remove one “as”.
  • Reply: done
  • In text “Several papers showed that biosurfactants produced by different bacteria exhibit several activities including antimicrobial, antiadhesive, antiviral, anticancer, anti-HIV, anti-inflammatory as well as immunomodulatory activities [8,13,18]. Remove last “activities”. Perhaps this paragraph corresponds more to the "Discussion" section.
  • Reply: The word has been removed. However, on our opinion this sentence can not be changed as part of discussion, because it contains information fundamental for background.
  • In text “Well-known hydrophobic peptide from  tendae…”. Change S from red to black font colour.
  • Reply: done
  • In text: “lipopeptides from ruberthat showed an antifungal activity”. Remove “an”.
  • Reply: done
  • In text: “peptides also exhibiting an antimicrobial activity”. Remove “an”.
  • Reply: done

Materials and Methods

  • In title 2.1: “Growth condition of the biosurfactant-producing bacterial strain”. Replace “condition” with “conditions”.

Reply: done

  • In text: “carried out by culturing of opacusR7 in a 3-L flask”. Remove “of”. Remove hyphen in 3-L.

Reply: done

  • In text: “However, the biosurfactant production…”. Remove “However, the”.

Reply: done

  • In text: “One hundred μL of each sample were…”. Replace “One hundred” with “100”.

Reply: done

  • In section 2.4: “3. Tests with ethanol-sulphuric acid, iodine vapor, UV, and cerium:molybdate reagent were performed on the purified biosurfactant to evaluate the presence of carbohydrates, unsaturated carbons, and oxidizable components, respectively. Each assay was carried out on TLC plates. The results were compared to the reaction of glucose and oleic acid to the same reagents.” The text that should go in this section is formatted as a new section title. This causes all subsequent numbering to be unbalanced. You should put this text as normal body text and correct the subsequent numbering.

Reply: The evidenced text has been incorrectly recognized as the title of a paragraph, while it is the content of the paragraph 2.4 with title “Determination of carbohydrates, unsaturated carbon chains, and oxidizable components of the biosurfactant compound”.

The numeration of the chapters has been revised.

  • In text: “biosurfactant at a concentration of 2-5 mg mL-1”. Does this mean that concentrations between 2 and 5 mg mL-1 were tested, or did it perhaps mean that a single concentration at 2.5 mg mL-1 was used? Change accordingly.

Reply: The text has been changed accordingly.

  • In text: “…ATCC 6538 were inoculated into the suitable medium”. Specify the culture medium used.

Reply: done

Results

  • In sections 4.4, 4.5 and 4.6 the first-person plural "we" is used repeatedly. These phrases should be changed to impersonal style.

Reply: all the sentences have been rephrased as suggested.

Discussion

  • In text: “Rhodococcus opacusstrain R7 is a…”. Replace “Rhodococcus with R.

 Reply: done

References

  • A full stop must be added after each reference.

Reply: The full stop has been added in PDF file from the Staff.

  • There are references, such as 15, 39 and 44, that have their doi underlined. Remove the underline.

Reply: done

  • References 2 and 31 do not have the year in bold.

Reply: done

Reviewer 2 Report

The manuscript doesn’t follow the journal instruction, the lines didn’t number, so the revision and the response to revision may be difficult many linguistic mistakes, so the manuscript needs to be extensively revised

Abstract

Replace “are suitable producers” with “excrete”

Replace “such as” with “which act as”

The surfactant, don’t bold

Introduction

The introduction is well and inclusive, however, many mistakes should be considered

Replace “moieties” with “residues”

also, in response to water-soluble” delete also

replace “a variety of chemical structures” with “several chemical groups”

replace “structurally related to peptides” with “belongs to peptides”

replace “In view of the antibiotic resistance increase evidenced by microorganisms and pathogens against the existing antimicrobial drugs” with “Because of increasing of antibiotic-resistant microorganisms, …

  1. tendae scientific name must be full

Materials and methods

Adjust the citations at the end of sentences

Provide model and origin of each device

Adjust the font style one bold and other fine and continually numbered the materials and methods sec as a second section not third

Results and discussion

Enhance figure 1 resolution

Add more recent discussion

Reduce conclusion

Author Response

Point-by-point

Reviewer 2

-The manuscript doesn’t follow the journal instruction, the lines didn’t number, so the revision and the response to revision may be difficult many linguistic mistakes, so the manuscript needs to be extensively revised

Reply: The manuscript has been completely revised and many mistakes corrected.

Abstract

-Replace “are suitable producers” with “excrete”

Reply: done

-Replace “such as” with “which act as”

Reply: done

- This sentence “The chemical and biochemical analyses of the isolated compound demonstrated that the biosurfactant could be classified as a hydrophobic peptide of 1292 MW possessing the following amino acid residues: Ala, Thr, Asp, Gly, Ser.” is not clear, please can you be more precise on the structure to explain the mass of 1292.

Reply: The sentence has been revised to be more clear.

- The text evidenced: The surfactant activity of this R. opacus R7 compound was quantified by the critical micelle dilution (CMD) method and the critical micelle concentration (CMC) was estimated around 20 mg L-1 with a corresponding surface tension of 48 mN m-1.

Reply: The evidenced text has been changed into “Surfactant activity of this R. opacus R7 compound was quantified by the critical micelle dilution (CMD) method and the critical micelle concentration (CMC) was estimated around 20 mg L-1 with a corresponding surface tension of 48 mN m-1.

Introduction

-Replace “moieties” with “residues”

Reply: done

-also, in response to water-soluble” delete also

Reply: done

-replace “a variety of chemical structures” with “several chemical groups”

Reply: done

-replace “structurally related to peptides” with “belongs to peptides”

Reply: done

replace “In view of the antibiotic resistance increase evidenced by microorganisms and pathogens against the existing antimicrobial drugs” with “Because of increasing of antibiotic-resistant microorganisms, …

Reply: done

- Please modify this sentence “The first detailed studies of biosurfactant antimicrobial properties commenced over the last two decades”, as the evidence for the antimicrobial activities of Bacillus subtilis lipopeptides is much older, see these paper for example

https://www.jstage.jst.go.jp/article/antibiotics1968/32/8/32_8_828/_article/-char/ja/

https://www.jstage.jst.go.jp/article/antibiotics1968/39/7/39_7_888/_article/-char/ja/

Reply: The meaning of the sentence was that the first studies regarding antimicrobial activity on compounds characterized as surfactants commenced later. Therefore, the sentence has been modified to be more clear.

- The text evidenced: the well-known hydrophobic peptide from S. tendae can be included in the streptofactin class.

Reply: The evidenced text has been modified into Streptomyces tendae.

Materials and methods

Adjust the citations at the end of sentences

Reply: done

Provide model and origin of each device

Reply: done

-Adjust the font style one bold and other fine and continually numbered the materials and methods sec as a second section not third

- Please correct numberring of the chapter

Reply: The numeration of the chapters has been revised.

-The text evidenced “3. Tests with ethanol-sulphuric acid, iodine vapor, UV, and cerium:molybdate reagent were performed on the purified biosurfactant to evaluate the presence of carbohydrates, unsaturated carbons, and oxidizable components, respectively. Each assay was carried out on TLC plates. The results were compared to the reaction of glucose and oleic acid to the same reagents.”:

Reply: The evidenced text has been incorrectly recognized as the title of a paragraph, while it is the content of the paragraph 2.4 with title “Determination of carbohydrates, unsaturated carbon chains, and oxidizable components of the biosurfactant compound”.

- name of the city

Reply: The name of the city has been added.

- Please be more specific about the protocol used for ESI-full Mass Spectrometry:

Reply: The protocol has been modified to be more specific as required by the Reviewer.

- Supplier, etc.. (the electrospray ionization ESI-full mass spectrometry on Q-TOF Synapt G2-Si mass spectrometer.)

Reply: The suppliers and the provenience were added.

- Please be more specific about the protocol used for the NMR Spectroscopy of the biosurfactant compound

Reply: The protocol has been modified to be more specific as required by the Reviewer.

- name of the city (Bruker Avance 500 MHz and Avance I 600 MHz spectrometers (Germany))

Reply: The name of the city has been added.

- What is the medium used to carried-out this test?

Reply: The medium used was Luria-Bertani (LB) as reported in the text.

- In this figure please add a "negative control" (Figure 1)

Reply: The control was added in the panel B of the Figure 1. So the Figure 1 has been changed.

- Please add "a negative control" in this figure (Figure 4)

Reply: The negative control is reported one time in the Figure 1.

- I suggest making a mix of figures 3 and 5

Reply: We appreciated the suggestion, but we think that the mixture of Figure 3 and 5 doesn’t seem coherent with the experiments respect to the text and easy to read. In any case, we can do it.

- The paragraph 3.4. needs to be completely revised, the title speaks of biochemical characterisation and the text of proteinase K and antibacterial activity.

Reply: In order to be clear we have changed the title of the paragraph as follow:

3.4. Effects of proteinase K enzyme on the isolated biosurfactant compound and total protein assay

Regarding the content of the text, we report in this paragraph the antimicrobial activity only to demonstrate the results of the proteinase K activity.

- Please add the corresponding mass spectrum in the supplementary data.

Would it be possible for you to carry out a more detailed structural study using, for example, LC-MS-MS to determine the precise amino acid sequence?

Reply: The corresponding mass spectrum was added in the Supplementary Data as required by the Reviewer as Figure S1.

Regarding the structural details, we thank the Reviewer for the suggestion; however, we think that the structural details would require other studies that are not feasible and obtainable on the bases of the conducted analyses.

- This figure (Figure 8) does not add anything to the previous one, please delete it and add to the discussion of figure 6

Reply: The Figure 8 was deleted as suggested by the Reviewer.

- What is the confidence interval and/or significance of these results, could you please complete (Table 1. MIC spectrum of the purified biosurfactant produced by R. opacus R7 against E. coli ATCC 29522 and S. aureus ATCC 6538.)

Reply: In order to answer to the Reviewer, we applied to our data a statistic test such as the t-student. The t-student test was applied for the growth inhibition of E. coli and S. aureus at MIC value and it demonstrated that our data are significant with a p-value < 0.05.

For the E. coli, two-tailed p-value was less than 0.0001 with 95% confidence interval of this difference from 87.57 to 116.43; for S. aureus, two-tailed p-value equals 0.0253 with 95% confidence interval of this difference from -174.38 to -19.62.

However, we have never seen reported these considerations related to this kind of experiments in other papers. In any case, if the Reviewer retains that this test is necessary, we can report the p-value in the Table.

- A real plus for your study would be to carry out an in-silico study of the production potential of this Rhodococcus opacus R7 strain using the Antismash software. I see that part of the genome is available on NCBI. I strongly encourage you to carry out this study to improve the quality of the analysis of your results

https://www.ncbi.nlm.nih.gov/assembly/GCF_000736435.1/

Reply: We had previously taken in consideration to carry out an in-silico study of the production potential of this Rhodococcus opacus R7 strain using the Antismash software as indicated also by the Reviewer; nevertheless the genome of R7 strain is the largest among rhodococci and considering the specificity of the novel biosurfactant that we identified in this study, the in-silico analysis turned out to be not exhaustive for the purpose of this article and we are not sure how much it could improve the results. A dedicated study should be carried out to perform a comprehensive in-silico prediction of the production potential of this strain.

In any case, we report here a summary of the main results obtained using Antismash software on R. opacus R7 genome:

The prediction of secondary metabolites was performed on R. opacus R7 genome using both the anti-SMASH and the anti-SMASH-incorporated NaPDoS pipelines (Blin et al. 2021). This analysis showed different biosynthetic gene clusters (BGCs) such as polyketide, terpene, saccharides, post-translationally modified peptides (RiPPs) and several non-ribosomal peptide synthetase (NRPS), classified in 32 known families.

In line with what was observed for other rhodococci, R. opacus R7 possesses a large number of putative BGCs with a ratio NRPS vs PKS (Polyketide synthase) higher compared to other actinomycetes and among PKS types, only PKS type I were observed. R. opacus R7 showed the highest number of gene clusters among 28 different strains including 20 strains belonging to Rhodococcus genus (Ceniceros et al. 2017).

NRPS are multi-modular enzyme complexes comprising activation or adenylation domain (A), modification or a peptidyl carrier protein (PCP) or thiolation domain (T), and condensation domain. All NRPS identified in R. opacus R7 showed a C domain subtype called the C-starter domain that is the first in the assembly domain lines. This starter domain drives the acylation of the first amino acid (with the presence of β -hydroxy or β-amino fatty acid) of the peptide moiety (Ceniceros et al. 2017).

-Add more recent discussion

Reply: Recent Discussion have been added.

Reduce conclusion

Reply: Conclusion has been reduced.

Round 2

Reviewer 2 Report

The authors have carefully processed all comments. The quality of the manuscript has increased significantly. I have no further comments.

Author Response

The Reviewer has written that the quality of the manuscript has been improved significantly.

Reply: Thank you for your comment.